# Intelligent Path Planning with an Improved Sparrow Search Algorithm for Workshop UAV Inspection

**DOI:** 10.3390/s24041104

**Published:** 2024-02-08

**Authors:** Jinwei Zhang, Xijing Zhu, Jing Li

**Affiliations:** 1School of Mechanical Engineering, North University of China, Taiyuan 030051, China; s202102022@st.nuc.edu.cn (J.Z.); li-jing@nuc.edu.cn (J.L.); 2Shanxi Provincial Key Laboratory of Advanced Manufacturing Technology, North University of China, Taiyuan 030051, China

**Keywords:** UAV, sparrow search algorithm, firefly algorithm, chaotic sequence, path planning

## Abstract

Intelligent workshop UAV inspection path planning is a typical indoor UAV path planning technology. The UAV can conduct intelligent inspection on each work area of the workshop to solve or provide timely feedback on problems in the work area. The sparrow search algorithm (SSA), as a novel swarm intelligence optimization algorithm, has been proven to have good optimization performance. However, the reduction in the SSA’s search capability in the middle or late stage of iterations reduces population diversity, leading to shortcomings of the algorithm, including low convergence speed, low solution accuracy and an increased risk of falling into local optima. To overcome these difficulties, an improved sparrow search algorithm (namely the chaotic mapping–firefly sparrow search algorithm (CFSSA)) is proposed by integrating chaotic cube mapping initialization, firefly algorithm disturbance search and tent chaos mapping perturbation search. First, chaotic cube mapping was used to initialize the population to improve the distribution quality and diversity of the population. Then, after the sparrow search, the firefly algorithm disturbance and tent chaos mapping perturbation were employed to update the positions of all individuals in the population to enable a full search of the algorithm in the solution space. This technique can effectively avoid falling into local optima and improve the convergence speed and solution accuracy. The simulation results showed that, compared with the traditional intelligent bionic algorithms, the optimized algorithm provided a greatly improved convergence capability. The feasibility of the proposed algorithm was validated with a final simulation test. Compared with other SSA optimization algorithms, the results show that the CFSSA has the best efficiency. In an inspection path planning problem, the CFSSA has its advantages and applicability and is an applicable algorithm compared to SSA optimization algorithms.

## 1. Introduction

### 1.1. UAV in Intelligent Manufacturing

With the rapid development of technologies, artificial intelligence and the Internet of Things have been widely applied. As the industrial revolution has developed from 1.0 to 4.0, traditional manufacturing workshops have been changing towards the direction of intelligent manufacturing. The concept of “intelligent manufacturing” was first proposed by American scholars Wright and Bourne and is defined as “the process of using integrated knowledge engineering, manufacturing software system and robot vision and other technologies to complete small batch production by intelligent robots alone without human intervention” [1]. Up to now, the specific meaning of “intelligent manufacturing” has been evolving, the academic circles have not yet reached a unified consensus. According to an earlier study by the US Department of Energy on the topic, intelligent manufacturing is a combination of technologies and practices of sensing, instrumentation, monitoring, control and process optimization. It integrates information and communication technologies with the manufacturing environment to achieve the real-time management of energy, productivity and cost in factories and enterprises [2]. Monitoring and control are particularly important parts of the intelligent manufacturing process. Therefore, it is of great practical importance to study the inspection of intelligent manufacturing workshops.

With the reduction in UAV costs and the broadening of their applications, UAVs serve as an important tool in not only the military but all aspects of life, such as agriculture [3], surveying and mapping [4], power inspection [5], package delivery [6] and so on. In the intelligent manufacturing industry, intelligent inspection can be carried out using UAV’s mobility to monitor manufacturing equipment and processes. After the acquired information is processed for optimization, feedback control can be conducted on the manufacturing equipment through the Internet of Things and artificial intelligence. UAV has a wide range of application prospects in the intelligent manufacturing industry.

### 1.2. UAV’s Path Planning

The path planning of a UAV is an important work in its function. UAV’s path planning capability directly reflects its operational and risk management capabilities. A simple path planning algorithm classification framework is shown in Figure 1.

In global algorithm path planning, the path in the known environment is planned. The environment information has been given and the information such as the location of obstacles does not change.

The classic A* Algorithm was first published in 1968 by Peter Hart et al. [7]. It can be considered as an extension of the Dijkstra algorithm. Due to the guidance of heuristic functions, the A* algorithm usually has better performance.

Tabu Search Algorithm (TSA) is an optimization algorithm proposed by Glover in 1986 [8]. It is a Neighborhood Search Algorithm (NSA) based on greedy thinking. However, the biggest drawback of NSA is that it is prone to falling into local optima. To solve this problem, a taboo table is introduced to form the TSA, which has strong global search optimization ability.

In local algorithm path planning, the path in an unknown environment is planned. Sensors and other tools are used to obtain the current local environment information. Based on this, path planning and obstacle avoidance are carried out.

In 1986, Khatib first proposed the artificial potential field algorithm and applied it to the field of path planning [9]. The convergence speed of the artificial potential field method is fast, and the obtained path has high reachability, which is very suitable for the planning tasks that require high real-time and security-of-path generation.

On the basis of the A* Algorithm, Ferguson et al. proposed a dynamic path planning algorithm, namely the D* algorithm, which is suitable for path planning in a dynamic environment [10]. The D* algorithm has excellent real-time performance, which makes it more suitable for dynamic path planning in complex environments. However, the D* algorithm usually searches feasible waypoints in a large space range, which makes the convergence speed of the algorithm unsatisfactory.

The intelligent bionic algorithm is a kind of random search method that simulates natural biological evolution or group social behavior. Because its solution does not depend on gradient information when solving, it is widely used in practical problems such as path planning. Intelligent bionic algorithms mainly include Ant Colony Optimization (ACO) [11], the Genetic Algorithm (GA) [12], Particle Swarm Optimization (PSO) [13] and so on. 

In this paper, we proposed an improved sparrow search algorithm: the CFSSA. The CFSSA is an intelligent bionic algorithm based on the sparrow search algorithm (SSA) [14] to enable UAVs to conduct optimal path planning during intelligent inspection.

## 2. Actual Path Planning Problem and Model Building

### 2.1. Problem Description

In an aircraft manufacturing workshop, to save manufacturing time, the workshop was divided into several areas according to different functions, including the parts area, laser area, assembly area and integration area. In this workshop, a UAV was used to conduct periodic cyclic intelligent inspection in each area. The internal structure of the manufacturing workshop is shown in Figure 2.

To better address the inspection path topic, the following assumptions were made after simplifications [15,16,17]:There was only one take-off/land location for UAV inspection in the aircraft manufacturing workshop. The UAV takes off and lands at the same location;There were five inspection areas in the aircraft manufacturing workshop: the parts area, laser area, assembly area, integration area and finished product area. Each inspection area had six inspection locations in sequence for a total of 30 inspection locations;One UAV was dispatched to carry out cyclic inspections of each inspection location; and the UAV only passed each inspection location once per inspection cycle.

It is assumed that $\;m=\left\{0,\;1,\;2,\;\ldots ,\;M\right\}$ where *m* represents the inspection locations. The conditions of *m* = 0 and *m* = *M* refer to the take-off and land location of the UAV. Other involved parameters are defined as follows:*t*_m_ is the time spent on each inspection location;*t*_max_ and *t*_min_ are the longest and shortest time spent on one inspection cycle, respectively;*d*_mn_ is the distance from location *m* to *n*;*L*_max_ is the maximum distance allowed for UAV inspection;*H*_max_ is the maximum flight height of the UAV in the aircraft manufacturing workshop;*V*_max_ is the maximum speed of the UAV during inspection; because the UAV needs to hover during inspection, there is no need to set a minimum speed;*V*_1_ is the flight speed of the UAV between inspection locations, which is assumed to be a constant;*V*_2_ is the flight speed of the UAV when at the inspection location;*h* is the flight height of the UAV during inspection;*D* is the entire distance covered by UAV during inspection.

The decision variables are defined as follows: (1)$$x_{mn}=\left\{\begin{array}{c} 1\;\;\;\;UAV\;flight\;from\;m\;to\;n\\ 0\;\;\;\;\;\;\;\;\;\;\;\;\;\;\;\;\;\;\;\;\;\;others \end{array}\right.$$
(2)$$y_{m}=\left\{\begin{array}{c} 1\;\;\;\;\;\;\;\;\;\;\;UAV\;passes\;an\;inspection\;location\\ 0\;\;\;\;UAV\;does\;not\;pass\;an\;inspection\;location \end{array}\right.$$

### 2.2. Mathematical Models

The optimization objective parameter is the shortest inspection path of the UAV in the workshop, given by
(3)$$minD=\overset{M}{\underset{m=1}{\sum }}\overset{N}{\underset{n=1}{\sum }}x_{mn}d_{mn}$$

The constraint condition of the maximum inspection speed of the UAV at the inspection location is given by
(4)$$V_{2}\leq V_{\max}$$

The constraint conditions of the maximum and minimum inspection time are given by
(5)$$\overset{M}{\underset{m=1}{\sum }}y_{m}t_{m}+\frac{L_{max}}{V_{1}}\leq T_{max}$$
(6)$$\overset{M}{\underset{m=1}{\sum }}y_{m}t_{m}+\frac{L_{max}}{V_{1}}\geq T_{min}$$

There is only one drone at each inspection location, expressed as
(7)$$\overset{M}{\underset{m=1}{\sum }}x_{mn}=y_{m}\;\;\;\;\forall m$$

The constraint condition of the maximum inspection distance during the UAV inspection is given by
(8)$$\overset{M}{\underset{m=1}{\sum }}\overset{M}{\underset{n=1}{\sum }}d_{mn}x_{mn}\leq L_{max}\;\;\;\;\;\forall m,n$$

### 2.3. Map Modelling

During the actual working process, we found that the UAV will directly climb to the working height in place after taking off. And the height remains unchanged throughout the inspection process. After the inspection is completed, the UAV flies directly above the take-off/land location and lands. Therefore, path planning during the inspection process can be regarded as a two-dimensional problem.

We modeled the map according to the real environment of the aircraft manufacturing workshop. Common maps include raster, topological and semantic maps. The raster map is simple in its data structure and can effectively express the variability of the space. And the two-dimensional raster map modelling method was used in this study to simulate the real UAV inspection environment in the workshop [15], as shown in Figure 3.

The colors of different raster areas in the constructed 2D raster map are defined as follows. The black parts are the fixed obstacles such as walls and wall columns. The pink parts are the work areas of the aircraft manufacturing workshop. The red parts are the UAV inspection areas. The default inspection area of the UAV is the range of a raster near the work area. The yellow part is the take-off and landing location of the UAV. The green part is the random positions of the staff.

Due to the flight control program settings of inspection, while inspecting a surrounded area, the UAV needs to move to the area’s center first, then fly around the area once and return to the area’s center. Therefore, the path planning problem can be transformed from moving between areas to moving between points, i.e., the travel salesman problem (TSP).

## 3. Principle of Standard SSA

### 3.1. Algorithm Introduction

The SSA, proposed by Xue et al. in 2020 [14], mimics the behavior of sparrow populations in the sense that the population is divided into producers and scroungers in the process of finding food. The producers are mainly responsible for finding food in a wide range of environments. They need to find the location and direction of food for the sparrow population. Conversely, the scroungers follow the producers to forage around the food location. Each individual sparrow’s role as a producer or scrounger is not fixed, and they need to be able to flexibly switch according to the situation.

In the natural environment, because the sparrows in the periphery are vulnerable to attack, they gradually adjust their positions to move closer to the center of the population to increase their security. Moreover, all sparrows are constantly on alert, and when one sparrow within the population detects danger, the entire population moves from its current location to a safe environment to continue foraging.

### 3.2. Algorithm Flow

When the SSA is started, it first initializes a group of random particles (random solutions) and sets the maximum number of iterations. It is assumed that there are *N* particles in a *D*-dimensional search space. Therefore, the population size of the particle swarm is *N.* Each particle has velocity and position properties.

Producer: The producer has a larger foraging search area than the scrounger because in addition to meeting its own food needs, it also needs to provide the direction of the foraging area for the entire population. During each iteration, the position of the *i*th producer particle is updated as follows:(9)$${x_{i}}^{k+1}=\left\{\begin{array}{c} {x_{i}}^{k}\cdot \exp\left(\frac{-i}{\alpha \cdot k_{max}}\right)\;\;\;\;if\;R<ST\\ {x_{i}}^{k}+Q\cdot L\;\;\;\;\;\;\;\;\;\;\;\;if\;R>ST \end{array}\right.$$
where *k* is the number of current iterations, *k*_max_ is the maximum number of iterations, *α* is a random number in the interval (0, 1), *Q* is a random number subject to the normal distribution, *L* is a matrix of 1 × *D* with all elements being 1, *R* is an alarm value in the range of [0, 1] and *ST* is a safety threshold in the range of [0.5, 1].

When *R* < *ST*, there is no danger in the vicinity, and the producer can search in the surrounding space. When *R* ≥ *ST*, the producer senses danger and moves in a random direction.

Scrounger: All particles except the producer are scroungers. They constantly monitor the producer’s information. Once the scrounger notices that the producer has found a better foraging area, they give up their current position and move to the better foraging area to look for food. During each iteration, the position of the *i*th scrounger particle (*x_i_*) is updated as follows:(10)$${x_{i}}^{k+1}=\left\{\begin{array}{c} Q\cdot \exp\left(\frac{{p_{w}}^{k}-{x_{i}}^{k}}{i^{2}}\right)\;\;\;\;\;\;\;\;\;\;\;\;\;\;\;\;if\;i>\frac{N}{2}\\ {p_{b}}^{k+1}+\left|\;{x_{i}}^{k}-{p_{b}}^{k+1}\;\right|\cdot A^{\prime }\cdot L\;\;\;\;\;\;\;if\;i\leq \frac{N}{2} \end{array}\right.$$
where *p*_b_ is the global best position, *p*_w_ is the global worst position and *A*’ is a matrix of *D* × *D*, with all elements being randomly 1 or −1.

When $i>\frac{N}{2}$, the amount of food obtained by the scrounger is too little, so it flies to other places to find food. When $i\leq \frac{N}{2}$, the scrounger follows the producer to the optimal foraging area.

Watchman: In the preset rules, each particle has a reconnaissance and early warning mechanism. A particle may be aware of danger and therefore abandon the current area and move to a safe area. A particle in this state is called a watchman. During each iteration, the position of the *i*th watchman particle is updated as follows:(11)$${x_{i}}^{k+1}=\left\{\begin{array}{c} {p_{b}}^{k}+\beta \cdot \left|\;{x_{i}}^{k}-{p_{b}}^{k}\;\right|\;\;\;\;\;\;\;\;\;\;if\;f_{i}>f_{b}\\ {p_{b}}^{k}+K\cdot \frac{{x_{i}}^{k}-{p_{w}}^{k}}{\left|\;f_{i}-f_{w}\;\right|+\varepsilon }\;\;\;\;\;\;\;\;if\;f_{i}=f_{b} \end{array}\right.$$
where *β* is a step size regulating factor, and its value is a random number with a normal distribution with the mean being 0 and the variance being 1. *K* is a random number within [−1, 1]. The parameter *f_i_* is the fitness of the *i*th particle, while *f*_b_ and *f*_w_ are the current best and worst fitness values, respectively. The parameter *ε* is a very small constant to prevent the denominator from being zero.

When $f_{i}>f_{b}$, the particle is located at the edge of the population, and it preferentially moves closer to the center of the population. When $f_{i}=f_{b}$, the particle is located in the middle of the population, and it moves randomly to get close to other particles to avoid being preyed upon.

The complete computation flow of the SSA is as follows:

Step 1: Initialize the population by setting up the population size *N*, the maximum number of iterations Iteration, the proportion of producers, the proportion of sparrows that are aware of danger and a safety threshold;

Step 2: Calculate the fitness of the current population’s individuals, and sort to find the current best and worst values;

Step 3: Select the particles with good fitness values as the producers according to the proportion of producers and update the producer’s position according to Equation (9);

Step 4: Treat the remaining particles in the population as the scroungers and update their positions according to Equation (10);

Step 5: Randomly select some individuals in the population as the particles that are aware of danger according to the proportion of watchmen and treat them as the watchmen. Update their positions according to Equation (11) and calculate the new fitness values. If the fitness is better than the current optimal value, update the positions;

Step 6: Calculate the fitness values and retain the position of the optimal individual;

Step 7: Check whether the stop criterion is met. If yes, stop the algorithm and output the optimal result. Otherwise, go to Step 2.

## 4. Optimization Design of the CFSSA

Similar to other swarm-based intelligence algorithms, the SSA also has defects in the optimization process, such as a high risk of falling into local optima and insufficient convergence accuracy, which need further research to improve. Many scholars have improved the SSA to address its shortcomings [18,19,20]. The efforts have prevented the SSA from falling into local optima to a certain extent and improved the performance of the algorithm. However, the SSA still needs to be further improved in terms of optimization accuracy, convergence speed and stability.

Based on the SSA, this paper proposes an improved algorithm that adds chaotic cube mapping initialization, a firefly algorithm disturbance search and a tent chaos mapping perturbation search. The proposed algorithm optimizes the initial population distribution through chaotic cube mapping initialization and uses the firefly algorithm disturbance strategy and tent chaos mapping perturbation strategy to further optimize and update the sparrow’s location after the sparrow search. It improves the search ability and enriches the diversity of solutions. Therefore, the algorithm can avoid falling into local optima.

### 4.1. Improvement of Population Initialisation Stage Based on Chaotic Cube Mapping

In the population initialization stage, the SSA relies on a simple random function. The particle population generated with such a mechanism has an uneven distribution and insufficient diversity. Therefore, the optimization ability of the algorithm fluctuates greatly due to the different locations of the randomly generated population, leading to insufficient stability of the algorithm. The convergence accuracy of the algorithm is also negatively affected. To solve this common problem in swarm-based intelligence algorithms, chaotic sequencing has been widely used. Chaotic sequencing can be used to generate the initial population of swarm-based intelligence algorithms and optimize the performance of mutants. According to different mathematical principles, chaotic sequencing can be achieved by mapping different chaotic models, including the widely used Tent-map, Logistics-map, Kent-map and Cube-map. Studies have shown that chaotic cube mapping offers better distribution uniformity and predictability than other mapping methods [21]. Therefore, chaotic cube mapping was adopted in this study to improve the initialization stage of the SSA.

Chaotic cube mapping is described by the following formula:(12)$$y_{i+1}=4{y_{i}}^{3}-3y_{i},-1<y_{i}<1,\;y_{i}\neq 0,\;i=0,1,\ldots ,N$$

The population initialization process of the SSA can be improved with chaotic cube mapping according to the following three steps.

First, assuming there are *N* particles in the *D*-dimensional search space, a random particle is first generated with each dimension having a range of [−1, 1], $Y=\left(y_{1},y_{2},y_{3},\;\ldots \;,y_{D}\right)$;

Second, each dimension of each particle is iterated with Equation (12) by a total of *N* − 1 iterations, generating *N* particles;

Finally, after the iterations of all particles are completed, they are mapped to the solution space of the problem according to Equation (13). Then, the position of each particle can be calculated by
(13)$$x_{i}=x_{min}+\frac{\left(x_{max}-x_{min}\right)\cdot \left(y_{i}+1\right)}{2}$$
where *x*_min_ and *x*_max_ represent the upper and lower limits of the solution space, respectively.

As the initial location information of the sparrow population, the data generated via chaotic cube mapping are more evenly distributed in the solution space. This approach can effectively improve population diversity, algorithm stability and global search capability. The population distribution generated via chaotic cube mapping is shown in Figure 4.

### 4.2. Improvement of Location Update Based on Firefly Algorithm Disturbance

The SSA may suffer from premature convergence when searching for the target in the solution space, resulting in insufficient accuracy of the optimal solution. The producer can easily jump directly to the area surrounding the current extreme value in the late stage of the position update iteration, leading to an insufficient search in the range and falling into local optima. The optimization accuracy is also affected.

To overcome this shortcoming, the iterative strategy of firefly algorithm disturbance was introduced after the sparrow search. The firefly disturbance strategy was applied to the algorithm. With the help of the firefly’s luminous attraction, the individuals with better positions in the neighborhood structure can be found to enhance the diversity of the solution.

The Firefly Algorithm (FA) was proposed by Yang, a Cambridge scholar [22], to achieve location optimization by simulating the luminous behavior of fireflies. The mathematical modelling in the FA is described below.

The relative fluorescence luminance *I* of the firefly is
(14)$$I=I_{0}\cdot exp\left(-\gamma r_{ij}\right)$$
where *I*_0_ is the maximum fluorescence brightness (i.e., the fluorescence brightness at zero distance). The individual having a better fitness value has a greater *I*_0_ value. The parameter *γ* is the light intensity absorption coefficient, reflecting the attenuation effects of increased distance and medium absorption on the firefly’s fluorescence. The parameter *r_ij_* is the spatial distance between fireflies *i* and *j*.

The relative attraction of the firefly *δ* is given by
(15)$$\delta =\delta_{0}\cdot exp\left({-\gamma r_{ij}}^{2}\right)$$
where *δ*_0_ is the maximum attraction (i.e., the attraction when the distance between two fireflies becomes zero).

The location update is calculated for the disturbance of firefly *i* caused by firefly *j* as follows:(16)$$x_{i}=x_{i}+\delta \cdot \left(x_{j}-x_{i}\right)+\zeta \cdot \left[rand\left(\cdot \right)-0.5\right]$$
where ζ is the step size regulating factor and rand(·) is a random factor subject to the uniform distribution with the range of [0, 1].

### 4.3. Improvement of Location Update Based on Tent Chaos Mapping Perturbation

In frequently used chaotic models, the traversal uniformity and convergence speed of tent chaos mapping are all preferable. Research proved that tent chaos mapping can be used as a chaotic sequence for generating optimization algorithms [23]. Therefore, tent chaos mapping perturbation was also applied to the algorithm. By adding tent chaos mapping perturbation terms, the search area can be expanded, and the global search ability of the algorithm can be improved.

Tent chaos mapping is described by the following formula: (17)$$y_{i+1}=\left\{\begin{array}{c} 2y_{i}+rand\left(0,\;1\right)\frac{1}{N},\;0\leq y_{i}\leq \frac{1}{2},\;i=0,1,\ldots ,N\\ 2\left(1-y_{i}\right)+rand\left(0,\;1\right)\frac{1}{N},\;\frac{1}{2}<y_{i}\leq 1,\;i=0,1,\ldots ,N \end{array}\right.$$

The formula after Bernoulli transformation is given by
(18)$$y_{i+1}=\left(2y_{i}\right)mod1+rand\left(0,\;1\right)\frac{1}{N},\;i=0,1,\ldots ,N$$

The population location update process of the SSA can be improved with tent cube mapping according to the following three steps.

First, generate chaotic variable $Y_{d}$ with Equation (18). $Y_{d}$ includes all particles which need to be tent chaos mapped;

Second, carry the chaotic variable into the solution space via
(19)$$y_{i}=x_{min}+\left(x_{max}-x_{min}\right)Y_{d}$$
where *x*_min_ and *x*_max_ represent the upper and lower limits of the solution space, respectively.

Finally, perform chaotic perturbation via
(20)$$x_{i}=\left(x_{i}+y_{i}\right)/2$$

Adding firefly algorithm disturbance and tent chaos mapping perturbation can help to prevent the solution from falling into local optima. Meanwhile, the sparrow position can be further updated to enhance the overall search of the population, improving the convergence accuracy.

### 4.4. Algorithm Flow of the CFSSA

The improved algorithm is called the chaotic mapping–firefly sparrow search algorithm (CFSSA). These improvements can enhance the stability, convergence accuracy and optimization accuracy of the original SSA.

The algorithm flow is given as follows:

Step 1: Initialize the population, and set the population size *N*, the maximum number of iterations Iteration, the proportion of producers, the proportion of sparrows that are aware of danger and the safety threshold. Use chaotic cube mapping to modify the initial positions of the particles to make their distributions more uniform and improve the diversity of the population;

Step 2: Calculate the fitness values of the individuals in the current population and sort them to find the current best value and worst value;

Step 3: Select the particles with better fitness values as the producers according to the proportion of producers and update their positions according to Equation (9);

Step 4: Treat the remaining particles in the population as the scroungers and update their positions according to Equation (10);

Step 5: Randomly select some individuals in the population as the particles that are aware of danger according to the proportion of watchmen and treat them as the watchmen. Update their positions according to Equation (11) and calculate the new fitness values. If the fitness is better than the current optimal value, update the positions;

Step 6: Check the particles’ fitness values and compare them with average fitness value $f_{avg}$. If $f_{i}<f_{avg}$, the particle is more inclined to gather together; this particle goes to Step 7. If $f_{i}\geq f_{avg}$, the particle is more inclined to scatter, this particle goes to Step 8;

Step 7: Introduce the FA where the particles in the population are equivalent to individual fireflies. According to the position after the previous iteration, calculate the fitness function value as the maximum fluorescence brightness of each firefly (*I*_0_). Calculate the relative fluorescence brightness *I* and the attraction *δ* of each firefly according to Equations (14) and (15). Then, determine the search direction of the population. Use Equation (16) of firefly disturbance to update the position of the population. Randomly disturb the fireflies in the optimal positions. Go to Step 9;

Step 8: Use tent cube mapping to modify the positions of the particles;

Step 9: Calculate the fitness values and retain the position of the optimal individual;

Step 10: Check whether the stop criterion is met. If yes, end the algorithm and output the optimal result. Otherwise, go to Step 2.

## 5. Reference Function Test of the CFSSA

To verify the optimization performance of the CFSSA, several different reference functions were selected for testing. There are 23 commonly used reference functions, and six representative ones were selected in this study [24,25,26]. The function parameters are shown in Table 1.

In frequently used intelligent bionic algorithms, the PSO [13], Whale Optimization Algorithm (WOA) [27] and SSA [14] were selected for a performance comparison with the CFSSA. The integrated development environment of the test was MATLAB R2020a. The operating system was WIN10 with 64 bits. All of the calculation processes were repeated 30 times, and the best, worst and average values and the standard deviation of the results were taken as the evaluation criteria of algorithm performance. The number of iterations of the four algorithms was set to 1000, and the population size was set to 100. The simulation results are shown in Table 2.

To intuitively illustrate the optimization effect of the algorithm, Figure 5 and Figure 6 show the search spaces of the reference functions and the convergence curves of the four algorithms. 

The results of the simulation test using the single-peak, multi-peak and low-dimensional multi-peak test functions reflect the performance of the four algorithms in terms of local search capability, global search capability and ability to jump out of local optima. With the single-peak test function $f_{1}\left(x\right)\sim f_{3}\left(x\right)$, the CFSSA showed good convergence speed and optimization accuracy. It could accurately find the optimal value, performing significantly better than the other algorithms. With the multi-peak test function $f_{4}\left(x\right)\sim f_{5}\left(x\right)$ and the low-dimensional multi-peak test function $f_{6}\left(x\right)$, the CFSSA could effectively escape local optima and find the global optimal solution. Although the improvement in the order of magnitude of the optimization result was usually small, the CFSSA exhibited the highest optimization accuracy. It could obtain a good average value with a small standard deviation. These results indicate that the CFSSA is robust and has significantly better optimization stability than the other algorithms.

## 6. Application of the CFSSA to Actual Path Planning Problem

### 6.1. Performance Comparison with Traditional Algorithms

The path planning problem’s description and math model was shown in Section 2. On this basis, we selected ACO, the GA, PSO, the SSA and the CFSSA for performance analysis. The number of iterations was set to 1000, and the population size was set to 100. Thirty simulation tests were conducted to compare the algorithms by using the 2D raster model map to generate optimal paths. The simulation results showed that the shortest inspection path of the UAV was 233.196 m and the shortest flight time was 15.019 s. The comparison is shown in Table 3, Table 4 and Table 5.

The following results can be obtained from the tables:

The CFSSA has the best optimization performance. The CFSSA found the global optimal path length solution of 233.196 m nine times. The GA found the global optimal solution two times. Other algorithms fell into local optima and could not find the global optimal solution. The average path length given via the CFSSA is 238.808 m, which is 3.20% higher than the SSA and even higher compared to the other algorithms.

The CFSSA has the best stability. The standard deviation of the CFSSA is 4.84% higher than the SSA and even higher compared to the other algorithms.

The CFSSA has the best efficiency. Because the numbers of iterations (and populations) of the algorithms used in the test were the same, the test time of the CFSSA and the SSA was very close. But compared with algorithms other than the SSA, the CFSSA has the shortest test time and the best efficiency. In addition, although the GA can find the global optimal solution, its test time is the longest among all algorithms and has the worst efficiency.

Figure 7 and Figure 8 show the optimal inspection paths with the SSA and CFSSA, respectively. The cyan part indicates the planned path. Figure 9 shows the comparison of the iteration effect before and after the CFSSA’s improvement. As shown, the CFSSA could iterate to the global optimal solution faster than the SSA, while the SSA was more likely to fall into local optima.

### 6.2. Performance Comparison with Other SSA Optimization Algorithms

We selected four SSA optimization algorithms proposed by other scholars to verify the performance differences between the CFSSA and other SSA optimization algorithms. These algorithms are called CSSA [18], MSISSA [19], QMESSA [20] and ICSSA [28]. Parameters of the math model and raster map were unchanged. And the comparison is shown in Table 6 and Table 7.

The following results can be obtained from the tables: 

The ICSSA has the best optimization performance. All of the algorithms could find the global optimal solution. They found the global optimal path length solution of 233.196 m 9 times, 12 times, 9 times, 11 times and 14 times, respectively. The average path length given via the ICSSA is 237.358 m, which is higher than the other algorithms.

The QMESSA has the best stability. The standard deviation of the QMESSA is higher than the other algorithms.

The CFSSA has the best efficiency. The CFSSA has the shortest test time and the best efficiency.

During the inspection process, something’s location in the workshop will change with time (e.g., the staff). For safety reasons, the UAV’s path needs to avoid being directly above them, which means the UAV may need to recalculate its subsequent path after passing a point. Therefore, the speed of path calculation is an important influencing factor. The CFSSA has the fastest calculation speed and the best efficiency, as well as preferable optimization performance and stability. In the SSA optimization algorithms, the CFSSA has its advantages and applicability for the inspection path planning problem.

## 7. Conclusions

An improved SSA was proposed by adding chaotic cube mapping initialization, firefly algorithm disturbance search and tent chaos mapping perturbation search. The initial population distribution was optimized by using chaotic cube initialization. Then, the locations of the sparrows were further optimized and updated by using the firefly disturbance strategy and tent chaos mapping perturbation after the sparrow search. The proposed algorithm improved the search capability and enhanced the diversity of solutions.

Six different reference functions were used to validate the optimization performance of the proposed CFSSA. Compared with the PSO, WOA and SSA, the CFSSA demonstrated better convergence speed and convergence accuracy. The performance of the ACO, GA, PSO, SSA and CFSSA was simulated and analyzed in an 2D raster simulation map from a practical application, and the CFSSA demonstrated better performance than the other algorithms. The experiment showed that the proposed algorithm significantly increased the convergence speed and generated more concise and smooth flight routes.

Also, the performance of the CFSSA and the other four SSA optimization algorithms was simulated. The CFSSA has the best efficiency, preferable optimization performance and stability. In the inspection path planning problem, the CFSSA has its advantages and applicability and is an applicable algorithm.

## Figures and Tables

**Figure 1 sensors-24-01104-f001:**
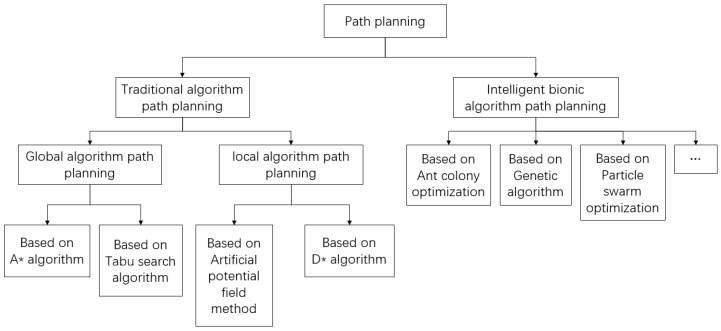
Path planning algorithm classification framework.

**Figure 2 sensors-24-01104-f002:**
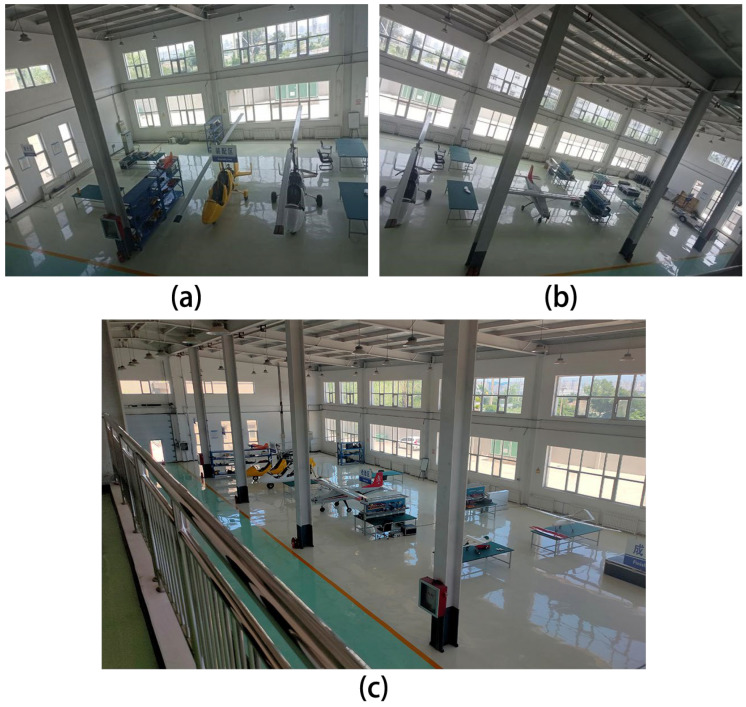
Photo of an aircraft manufacturing workshop: (**a**) left half part; (**b**) right half part and (**c**) entire workshop.

**Figure 3 sensors-24-01104-f003:**
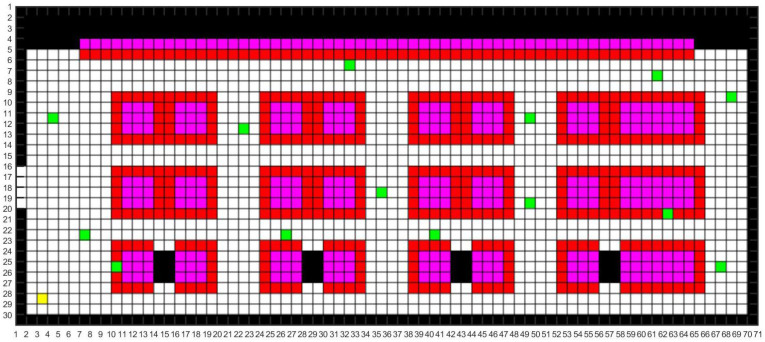
Two-dimensional raster map.

**Figure 4 sensors-24-01104-f004:**
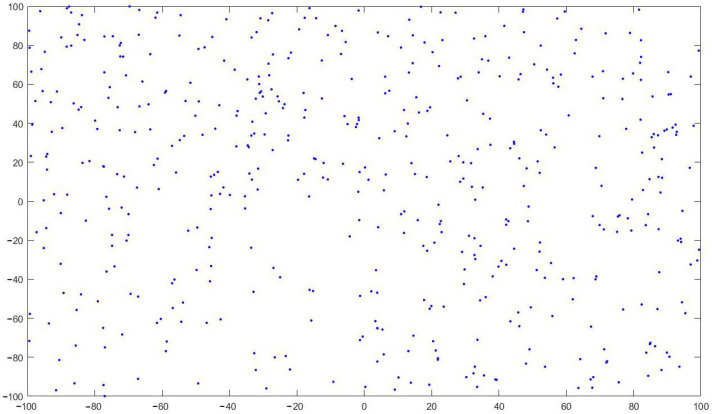
Population distribution map of chaotic cubic mapping.

**Figure 5 sensors-24-01104-f005:**
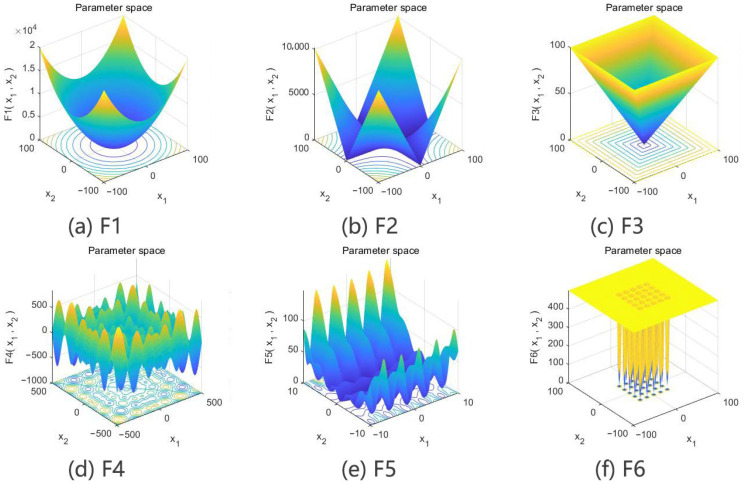
Function search spaces.

**Figure 6 sensors-24-01104-f006:**
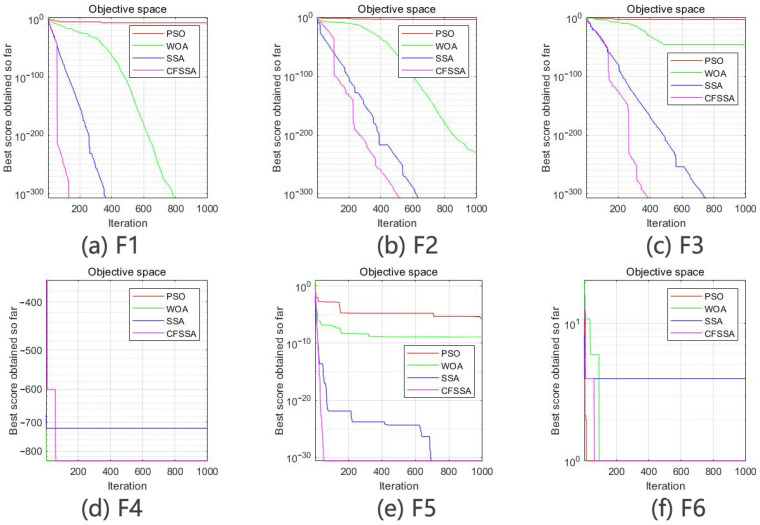
Function convergence curves.

**Figure 7 sensors-24-01104-f007:**
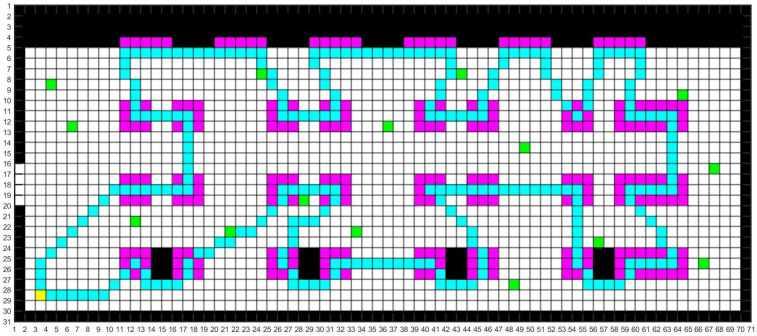
Optimal inspection path with the SSA.

**Figure 8 sensors-24-01104-f008:**
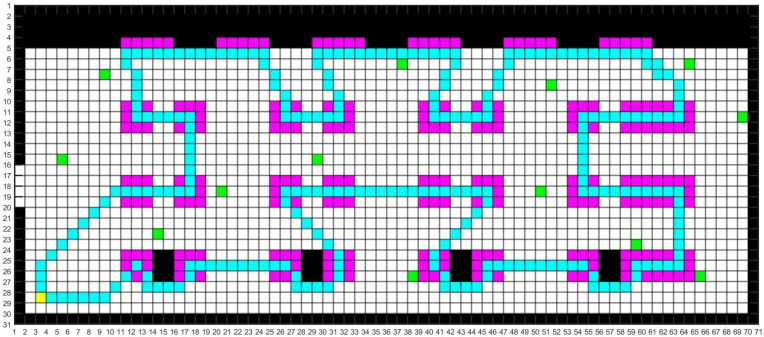
Optimal inspection path with the CFSSA.

**Figure 9 sensors-24-01104-f009:**
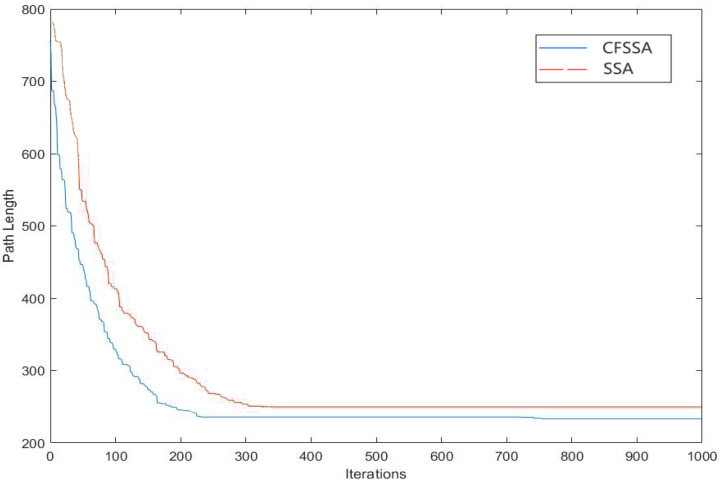
Comparison of iteration effect before and after improvement.

**Table 1 sensors-24-01104-t001:** Parameters of reference functions.

Reference Function	Range	Dimension	Minimum
$f_{1}\left(x\right)=\overset{n}{\underset{i=1}{\sum }}x_{i^{2}}$	[−100, 100]	30	0
$f_{2}\left(x\right)=\overset{n}{\underset{i=1}{\sum }}\left|x_{i}\right|+\overset{n}{\underset{i=1}{\prod }}\left|x_{i}\right|$	[−10, 10]	30	0
$f_{3}\left(x\right)=max_{i}\left\{\left|x_{i}\right|,\;1\leq i\leq n\right\}$	[−100, 100]	30	0
$f_{4}\left(x\right)=\overset{n}{\underset{i=1}{\sum }}-x_{i}sin\sqrt{\left|x_{i}\right|}$	[−500, 500]	30	418.982*n*
$f_{5}\left(x\right)=\frac{\pi }{n}\left\{10\sin\left(\pi y_{1}\right)+\overset{n-1}{\underset{i=1}{\sum }}{\left(y_{i}-1\right)}^{2}\left[1+10sin^{2}\left(\pi y_{i+1}\right)\right]+{\left(y_{n}-1\right)}^{2}\right\}+\overset{n}{\underset{i=1}{\sum }}u\left(x_{i},10,100,4\right)$ $y_{i}=1+\frac{x_{i}+1}{4}u\left(x_{i},a,k,m\right)=\left\{\begin{array}{c} k{\left(x_{i}-a\right)}^{m},x_{i}>a\\ 0,-a<x_{i}<a\\ k{\left(-x_{i}-a\right)}^{m},x_{i}<a \end{array}\right.$	[−50, 50]	30	0
$f_{6}\left(x\right)={\left(\frac{1}{500}+\overset{25}{\underset{j=1}{\sum }}\frac{1}{j+\sum_{i=1}^{2}{\left(x_{i}-a_{ij}\right)}^{6}}\right)}^{-1}$	[−65, 65]	2	1

**Table 2 sensors-24-01104-t002:** Simulation results.

Reference Function	Algorithm	Best	Worst	Average	Standard Deviation
$f_{1}\left(x\right)$	PSO	0.493	3.887	2.021	0.779
	WOA	3.986 × 10^−206^	3.658 × 10^−189^	1.231 × 10^−190^	0
	SSA	0	0	0	0
	CFSSA	0	0	0	0
$f_{2}\left(x\right)$	PSO	4.636 × 10^−2^	30.279	6.624	8.386
	WOA	1.074 × 10^−121^	1.155 × 10^−112^	6.398 × 10^−114^	2.272 × 10^−113^
	SSA	0	2.075 × 10^−274^	7.410 × 10^−276^	0
	CFSSA	0	1.032 × 10^−265^	3.691 × 10^−267^	0
$f_{3}\left(x\right)$	PSO	2.047	7.668	3.459	1.248
	WOA	6.110 × 10^−11^	71.588	16.367	20.591
	SSA	0	7.305 × 10^−217^	2.519 × 10^−218^	0
	CFSSA	0	4.290 × 10^−200^	1.483 × 10^−201^	0
$f_{4}\left(x\right)$	PSO	−8067.380	−5422.630	−6604.396	595.849
	WOA	−12,569.465	−8256.652	−12,030.776	928.457
	SSA	−12,213.757	−6152.891	−8920.496	2322.395
	CFSSA	−12,571.034	−11,402.927	−11,981.637	501.026
$f_{5}\left(x\right)$	PSO	0.982	8.266	3.186	1.527
	WOA	1.127 × 10^−5^	1.020 × 10^−4^	4.701 × 10^−5^	2.159 × 10^−5^
	SSA	1.432 × 10^−8^	6.362 × 10^−7^	1.836 × 10^−7^	1.398 × 10^−7^
	CFSSA	9.415 × 10^−9^	5.787 × 10^−7^	1.603 × 10^−7^	1.291 × 10^−7^
$f_{6}\left(x\right)$	PSO	0.998	0.998	0.998	0
	WOA	0.998	2.982	1.064	0.356
	SSA	0.998	12.670	6.953	4.841
	CFSSA	0.998	0.998	0.998	0

**Table 3 sensors-24-01104-t003:** Comparison of path length simulation data.

Test	ACO Path (m)	GA Path (m)	PSO Path (m)	SSA Path (m)	CFSSA Path (m)
1	268.367	255.708	253.176	253.176	245.577
2	270.510	265.546	253.137	248.174	242.989
3	253.145	243.409	243.409	243.409	243.501
4	256.099	261.024	246.249	246.249	237.854
5	268.435	245.644	248.176	253.241	233.196
6	258.645	253.719	248.792	246.329	241.926
7	257.755	277.582	247.841	247.841	241.011
8	260.239	243.214	248.078	243.214	235.561
9	265.013	235.567	242.928	245.382	238.080
10	256.872	256.872	235.208	237.844	233.196
11	271.006	286.492	255.520	258.101	233.196
12	258.774	266.238	246.333	248.821	238.189
13	270.813	273.275	246.194	246.194	238.123
14	264.964	282.629	254.870	252.347	241.357
15	271.396	258.946	244.007	248.987	241.581
16	254.429	247.159	239.890	242.313	247.556
17	263.022	250.845	248.410	243.539	243.487
18	262.847	253.021	245.651	245.651	233.196
19	256.624	233.196	244.519	242.098	233.196
20	266.101	268.520	244.329	241.910	237.611
21	260.450	240.794	245.708	245.708	242.298
22	254.012	249.264	239.768	237.394	233.196
23	258.212	260.626	246.146	241.320	244.086
24	268.271	255.735	253.228	250.721	238.503
25	267.142	272.043	242.633	245.084	244.621
26	265.271	270.276	247.753	250.256	233.196
27	272.043	272.043	252.077	249.581	240.433
28	255.531	233.196	240.929	243.363	233.196
29	268.906	273.886	253.967	248.987	241.130
30	278.617	255.821	248.222	253.288	233.196

**Table 4 sensors-24-01104-t004:** Algorithm path lengths for CFSSA with traditional algorithms.

Algorithm	Best	Worst	Average	Standard Deviation
ACO	253.145	278.617	263.450	6.498
GA	233.196	286.492	258.076	14.146
PSO	235.208	255.520	246.905	4.793
SSA	237.394	258.101	246.684	4.669
CFSSA	233.196	247.556	238.808	4.443

**Table 5 sensors-24-01104-t005:** Algorithm time costs for CFSSA with traditional algorithms.

Algorithm	Shortest Time Cost(s)	Longest Time Cost(s)	Average Time Cost(s)
ACO	21.037	24.890	22.986
GA	38.927	52.106	46.200
PSO	18.035	21.208	19.633
SSA	15.022	15.834	15.410
CFSSA	15.019	15.773	15.362

**Table 6 sensors-24-01104-t006:** Algorithm path lengths for CFSSA with other SSA optimization algorithms.

Algorithm	Best	Worst	Average	Standard Deviation
CFSSA	233.196	247.556	238.808	4.443
CSSA	233.196	241.683	237.736	4.105
MSISSA	233.196	249.094	240.134	6.580
QMESSA	233.196	243.170	238.285	3.928
ICSSA	233.196	244.329	237.358	4.256

**Table 7 sensors-24-01104-t007:** Algorithm time costs for CFSSA with other SSA optimization algorithms.

Algorithm	Shortest Time Cost(s)	Longest Time Cost(s)	Average Time Cost(s)
CFSSA	15.019	15.773	15.362
CSSA	36.247	39.504	37.885
MSISSA	18.657	19.493	19.015
QMESSA	22.381	25.548	23.914
ICSSA	17.200	19.034	18.146

## Data Availability

Data are contained within the article.

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
