# Peer review of "Intelligent Path Planning with an Improved Sparrow Search Algorithm for Workshop UAV Inspection"

_sensors, 2024, doi:10.3390/s24041104_

Round 1

Reviewer 1 Report

Comments and Suggestions for Authors

The manuscript studies the UAV path planning for intelligent workshop inspection based on an improved sparrow search algorithm. The investigated topic is interesting and the manuscript has certain contributions. The paper can be refined by considering the following issues.  

 1. Introduction can be refined into two parts. First, more relevant studies on path planning need to be introduced, such as the commonly used path planning algorithms Dijkstra's algorithm used in ‘Distributed multi-vehicle task assignment in a time-invariant drift field with obstacles (2019). Secondly, when introducing the usage of UAVs in various fields, another popular application scenario is the package delivery in logistics.

 2. The main contributions of the manuscript need to be clarified in Introduction.

 3. It is suggested to put Problem description and formulation section before the algorithm design section.

 4. Figure 2 is not clear.

 5. In simulations, more classical path planning algorithms are suggested to be compared with the designed method. Furthermore, the computational running time of the algorithms corresponding in Table 3 is suggested to be given for a fair comparison. 

Comments on the Quality of English Language

The quality of the English language of the manuscript is fine. 

Author Response

  1. The Introduction has been refined into two parts. Section 1.2 introduces UAV's path planning, including traditional algorithm path planning and intelligent bionic algorithm path planning.
  2. Author contributions has been included in the journal's template (Patents section). So it was removed from Introduction to avoid duplication.
  3. Problem description and formulation section has been moved to section 2. Then is the algorithm design section.
  4. The figure has been divided into two parts and replaced with a higher resolution version, which is numbered Figure 5 and Figure 6 in the revised manuscript.
  5. In simulations, we add Ant Colony Optimization (ACO), Genetic Algorithm (GA) and Particle Swarm Optimization (PSO) to compare with CFSSA, shown in section 6.1 in the revised manuscript. Also, the algorithms' time costs has been added, which is numbered Table 5 and Table 7 in the revised manuscript.

The revised manuscript is in the attachment. All changes are highlighted in yellow.

Reviewer 2 Report

Comments and Suggestions for Authors

This paper investigates intelligent workshop UAV inspection path planning algorithm based on improved sparrow search algorithm.

l  The author's discussion of relevant background is thorough in the introduction, but there is a small problem. The author has less discussion on the inspection path planning methods in the introduction. It is better to further elaborate in the introduction to make it easier for readers to read.

l  Figure 1 is not clear in the manuscript. Please modify it.

l  According to the content in Section 3, as shown in references 7-9, many scholars have made some improvements to SSA. Could you please add some relevant simulations and data in this paper to explain the advantages of the CFSSA with those?

l  Compare Figure 4., why is a part of the pink area missing from Figure 5? Does that mean these areas don't need to be inspected?

l  In Section 5.4, the author states that due to the same number of iterations and populations, the test time of CFSSA (the improved sparrow search algorithm in this paper) is very close to SSA. Is it possible to add some data and calculations in this paper to show that the efficiency of CFSSA is higher than that of SSA in the same time?

Comments on the Quality of English Language

Some of the expressions are not professional and concise enough. It is recommended to carefully check and improve the writing quality.

Author Response

  1. The Introduction has been refined into two parts. Section 1.2 introduces UAV's path planning, including traditional algorithm path planning and intelligent bionic algorithm path planning.
  2. The figure has been replaced with a higher resolution version, which is numbered Figure 4 in the revised manuscript.
  3. In simulations, we add these three improved algorithms from references to compare with CFSSA, shown in section 6.2 in the revised manuscript.
  4. Due to the flight control program settings of inspection in our research group, while inspecting a surrounded area, the UAV needs to move to the area's center first, then fly around the area once and return to the area's center at last. Therefore, the path planning problem can be transformed from moving between areas to moving between points. i.e. the travel salesman problem (TSP). Under this condition, the surrounded areas can be removed and only to consider center areas.
  5. The algorithms' time costs has been added, which is numbered Table 5 and Table 7 in the revised manuscript. The time cost of CFSSA is indeed very close to SSA, and there is no significant efficiency improvement compared with them. But we also adds some classical path planning algorithms to compare, including Ant Colony Optimization (ACO), Genetic Algorithm (GA) and Particle Swarm Optimization (PSO). Compared with these three algorithms, CFSSA has a significantly higher efficiency.

The revised manuscript is in the attachment. All changes are highlighted in yellow.

Round 2

Reviewer 2 Report

Comments and Suggestions for Authors

There are no other issues that need to be modified, except for some of the writing and expression that is not standardized

Comments on the Quality of English Language

There are no other issues that need to be modified, except for some of the writing and expression that is not standardized

Author Response

Thanks for your work. Unstandardized writing and expression will be revised uniformly before online publication.